# Maternal COVID-19 Serological Changes—Comparison between Seroconversion Rate in First and Third Trimesters of Pregnancy and Subsequent Obstetric Complications: A Cohort Study

**DOI:** 10.3390/v15122386

**Published:** 2023-12-05

**Authors:** Maria N. Rayo, Adriana Aquise, Irene Fernandez-Buhigas, Lorena Gonzalez-Gea, Coral Garcia-Gonzalez, Mirian Sanchez-Tudela, Miguel Rodriguez-Fernandez, Diego Tuñon-Le Poultel, Belen Santacruz, Maria M. Gil

**Affiliations:** 1Department of Obstetrics and Gynecology, Hospital Universitario de Torrejón, Torrejón de Ardoz, 28850 Madrid, Spain; nrayo@torrejonsalud.com (M.N.R.); ifbuhigas@torrejonsalud.com (I.F.-B.); lgonzalez@torrejonsalud.com (L.G.-G.); cggonzalez@torrejonsalud.com (C.G.-G.); miriansancheztudela@gmail.com (M.S.-T.); mrodriguez@torrejonsalud.com (M.R.-F.); bsantacruz@torrejonsalud.com (B.S.); 2School of Medicine, Universidad Francisco de Vitoria, Carretera Pozuelo a Majadahonda, Km 1.800, Pozuelo de Alarcón, 28223 Madrid, Spain; 3Synlab Laboratory, 28001 Madrid, Spain; diego.tunon@synlab.es

**Keywords:** SARS-CoV-2, pregnancy, morbidity

## Abstract

Pregnant women are especially vulnerable to respiratory diseases. We aimed to study seroconversion rates during pregnancy in a cohort of consecutive pregnancies tested in the first and third trimesters and to compare the maternal and obstetric complications in the women who seroconverted in the first trimester and those who did so in the third. This was an observational cohort study carried out at the Hospital Universitario de Torrejón, in Madrid, Spain, during the first peak of the COVID-19 pandemic. All consecutive singleton pregnancies with a viable fetus attending their 11–13-week scan between 1 January and 15 May 2020 were included and seropositive women for SARS-CoV2 were monthly follow up until delivery. Antibodies against SARS-CoV-2 (IgA and IgG) were analyzed on stored serum samples obtained from first- and third-trimester routine antenatal bloods in 470 pregnant women. Antibodies against SARS-CoV-2 were detected in 31 (6.6%) women in the first trimester and in 66 (14.0%) in the third trimester, including 48 (10.2%) that were negative in the first trimester (seroconversion during pregnancy). Although the rate of infection was significantly higher in the third versus the first trimester (*p* = 0.003), no significant differences in maternal or obstetric complications were observed in women testing positive in the first versus the third trimester.

## 1. Introduction

In early December 2019, a cluster of individuals suffering from pneumonia of unknown cause were identified in the city of Wuhan, Hubei Province, China. On 31 December 2019, the World Health Organization (WHO) was notified of these cases. Subsequently, the disease named COVID-19, a severe acute respiratory syndrome caused by the severe acute respiratory syndrome coronavirus 2 (SARS-CoV2) [1], has spread rapidly throughout most countries in the world. On 11 March 2020, the WHO declared a global pandemic emergency due to COVID-19. Since the outset of the pandemic, the gold standard for the diagnosis of active SARS-CoV2 infection remains real-time reverse-transcription polymerase chain reaction (rRT-PCR), a technique that detects viral RNA in the nasopharynx. However, false negatives have been reported for this technique, mainly due to problems related to sample collection and/or detection methods [2]. As a result, these limitations of the technique have led to some concerns as to whether rRT-PCR should be the gold standard test for the diagnosis of SARS-CoV2 infection. Conversely, serological tests bypass these technical challenges and, unlike rRT-PCR, are often faster, more cost-effective, user-friendly, and capable of identifying asymptomatic cases. As a result, they emerge as a valuable tool in gauging the pandemic’s scope [2]. On the basis of these advantages, serological surveys have been suggested as a complementary approach to RT-PCR to improve its sensitivity and provide rapid identification, study, and isolation of infected people and their contacts to prevent the spread of the coronavirus.

Research on previous pandemics caused by seasonal influenza, Middle East respiratory syndrome (MERS), and SARS-CoV1 have provided valuable information to face this novel mutation of Coronavirus, labeled as SARS-CoV2. Data from the past two decades reveal that over 10,000 individuals have affected by SARS-CoV1 and MERS-CoV infections, with respective mortality rates of 10.5% and 34.4%. More importantly, these pandemics have demonstrated that pregnant women had a higher vulnerability to respiratory infections due to physiological changes in their immune and cardiopulmonary systems and that a different obstetric impact was observed according to the trimester in which the infection was acquired [3,4,5]. Unfortunately, a challenging problem arises in the attempt to determine the impact of this novel mutation of Coronavirus, labeled as SARS-CoV2. Despite the increasing number of published studies, the reported data are still insufficient to draw definite and unbiased conclusions regarding the impact of SARS-CoV2 infection on obstetric morbidity or the clinical relevance of the time at which the infection occurs. Hence, the testing of specific SARS-CoV2 antibodies have emerged as a potential solution to address this challenge more effectively. Conducting sequential serological tests during the first and third trimesters of pregnancy could serve as a valuable clinical approach. This method could reliably pinpoint the timing of infection and accurately assess the impact of COVID-19 on pregnancy based on the trimester in which the woman was infected. Moreover, understanding the SARS-CoV2 immune status of women among the pregnancy presents e a unique opportunity to determine a more precise incidence of SARS-CoV2 infection. This knowledge facilitates a comprehensive follow-up, enabling the prompt detection of any possible complications. It also ensures high-quality assessment and healthcare for these pregnancies.

In this study, we aimed to assess the immune status of a complete and consecutive cohort of pregnant women throughout the pregnancy (from the first to the third trimester) covering the first (between March and June 2020) and second (between June and December 2020) waves of the COVID-19 pandemic [6] in one of the hotspots of Madrid (Spain), as Torrejón de Ardoz was one of the first places where population suffered from the infection. In addition, we aimed to analyze the rates of obstetric complications in the group of women who got infected by this new Coronavirus in the first trimester compared to those who seroconverted in the second or third trimesters of pregnancy.

## 2. Materials and Methods

### 2.1. Study Design and Population

This was a longitudinal, observational, and ambispective study carried out between 1 January and 25 December 2020, at the Hospital Universitario de Torrejón (HUT), Madrid, Spain, as part of the PRECORSE study (Study for PREgnancy CORonavirus Serologic Evidence), as has been previously described [7].

In our center, a surplus of antenatal blood samples from all pregnant women is routinely frozen and stored at −80 °C degrees at the Biobank Network of the Region of Murcia (Spain), BIOBANC-MUR (reg. number: B.0000859) for clinical and for research purposes. After the COVID-19 pandemic outbreak, all available stored serum samples collected during the first-trimester routine analysis between 1 January and 15 May 2020 were identified. The samples corresponding to women who gave their written informed consent to participate in this study and fulfilled the inclusion criteria (women over 18 years old, having singleton pregnancies with a nonmalformed life fetus, and having their pregnancy care in our Obstetric Unit), were retrieved from the freezers and transferred on dry ice to Synlab laboratory in Madrid, Spain, for determination of anti-SARS-CoV2 immunoglobulin A (IgA) and immunoglobulin G (IgG). These women were followed-up throughout their pregnancy according to the local protocol and, those testing positive in the first trimester were contacted and had monthly follow-ups in a specific clinic for maternal and fetal wellbeing and fetal biometry assessments. The surplus from their third-trimester routine blood was also tested for anti-SARS-CoV2 IgA and IgG antibodies. For every woman participating in our study, demographic characteristics, including age, ethnicity, body mass index, parity, smoking habits, medical disorders, and even blood type data were prospectively and thoroughly recorded at all hospital appointments throughout the pregnancy, until the last pregnant woman gave birth on 25 December 2020. All of the pregnant women included were classified according to their serological status in the first and third trimesters of pregnancy: those who were IgA or IgG anti-SARS-CoV2 positive in the first trimester of pregnancy (“positive serology 1T”); and those who were IgA or IgG anti-SARS-CoV2 positive in their third trimester, with a prior negative serology in the first trimester (“positive serology 3T”). Information about pregnancy outcomes (gestational hypertension, preeclampsia, gestational diabetes, fetal growth disorders, fetal anomalies, and other obstetrics complications, such as intrahepatic cholestasis, Rh isoimmunization, preterm birth, and shortened cervix) was meticulously collected from the hospital medical records and also by telephone interview if needed.

It is important to highlight that none of the participants had been vaccinated against SARS-CoV2, since the vaccine hadn’t yet been developed and this infection marked their first known encounter with SARS-CoV-2.

The Strengthening the Reporting of Observational Studies in Epidemiology (STROBE) Statement was used for reporting the results obtained in this study.

### 2.2. Laboratory Analysis and Interpretation

Determination of anti-SARS-CoV2 IgA and IgG was performed with an enzyme-linked immunosorbent assay (ELISA), providing semi-quantitative (extinction of the control patient sample/extinction of calibrator) serology results against the S1 domain of the spike protein of SARS-CoV2 in serum samples (Anti-SARS-CoV2 ELISA IgG and Anti-SARS-CoV2 ELISA IgA, Euroimmunn Medizinische Labordiagnostika AG, Lubeck, Germany). IgA and IgG were considered positive, indeterminate, and negative when the results were >1.1, 0.8 to 1.1, and <0.8, respectively, as recommended by the manufacturer (Appendix A).

For anti-SARS-CoV2 IgG, sensitivity and specificity reported by manufacturers is 83.3% and 95.0% respectively in confirmed COVID-19 cases and, 70.8% and 96.6% respectively in suspected COVID-19 cases [8]. Overall sensibility and specificity reported for anti-SARS-CoV2 IgA are 86.7% and 82.7%, respectively [9].

### 2.3. Statistical Analysis and Data Management

The data are expressed as the median (interquartile range) for continuous variables and in proportions (absolute and relative frequencies) for categorical variables. The Mann–Whitney test and Fisher’s exact test were used for comparing outcome groups for continuous and categorical data, respectively. The level of significance was set at 0.05. The statistical software package R was used for the data analyses [10], as well as table1 package [11].

### 2.4. Ethical Considerations

Approval from the local Research Ethics Committee Committee (Comité Ético de Investigación con Medicamentos de los Hospitales Universitarios Torrevieja y Elche-Vinalopó, No. Reg: 2020.028) was obtained prior to the start of the study. Signed informed consent was obtained from all pregnant women participating.

## 3. Results

### 3.1. Results from the First Trimester of Pregnancy

The surplus of routine first-trimester blood samples from 503 pregnant women was identified between 1 January and 15 May 2020, in the Hospital Universitario Torrejón in Madrid, Spain. Four hundred and eighty of the women were eligible, agreed, gave their consent to participate in the study, and had their blood samples from rutinary gestational analysis tested for anti-SARS-CoV2-specific antibodies. A total of 10 of the 480 pregnant women were excluded because of an insufficient amount of sample for analysis (*n* = 4) or lost to follow-up very early in their pregnancy (*n* = 6).

Finally, blood samples from 470 women were obtained (Table 1), including 31 (6.6%) samples that tested positive for SARS-CoV2 antibodies, either IgA, IgG, or both.

### 3.2. Results from the Third Trimester of Pregnancy

Of the 470 women with results from the first trimester testing, 7 had an early miscarriage (including 1 positive case in the first trimester), 4 had a late miscarriage, 3 terminated the pregnancy, and 54 were lost to follow-up (including 3 positive cases in the first trimester). Therefore, 402 samples were available for anti-SARS-CoV2-specific antibody testing in the third trimester, including 27 cases that were positive in the first trimester. A total of 66 (16.4%) of the 402 third-trimester samples tested positive, including 18 that had a positive result in the first trimester and 336 (83.6%) tested negative, including 9 that had a positive result in the first trimester. Therefore, SARS-CoV2 seroconversion during pregnancy occurred in 48 cases with complete follow-up (48, 12.8%, of the 375 negative pregnancies in the first trimester) (Figure 1 and Figure 2), which is statistically significantly higher than the seroconversion rate in the first trimester (*p* = 0.003).

### 3.3. Persistence of Antibodies among the Pregnancy

Of the 31 women with a positive serological test in the first trimester, 27 had their third-trimester blood samples tested for COVID-19 (1 had an early miscarriage and 3 were lost to follow-up). In addition, 18 (66.7%) of the 27 cases with complete follow-up still had a positive anti-SARS-CoV2 serology in the third trimester, while 9 of them (33.3%) had both negative IgG and IgA anti-SARS-CoV2 (Appendix A).

### 3.4. Maternal Morbidity

Among the positive cases, there were no differences in baselines characteristics between pregnant women who had a positive serology in the first trimester (*n* = 31) and those who seroconverted in the third trimester of their pregnancy (*n* = 48) (Table 1).

The present study showed no statistically significant differences when comparing the maternal or obstetric morbidity (gestational hypertension, preeclampsia, gestational diabetes, fetal growth disorders, fetal anomalies, and other obstetrics complications, such as intrahepatic cholestasis, Rh isoimmunization, preterm birth, and shortened cervix) according to the trimester in which SARS-CoV2 seroconversion occurred (Table 2).

## 4. Discussion

### 4.1. Main Findings of the Study

The main finding of this study is that, during the first COVID-19 pandemic peak, the seroconversion rate in the third trimester (12.8%) was double that in the first trimester (6.6%). However, obstetric or maternal complications did not differ between both groups. In addition, we demonstrated that about two-thirds (18/27) of the women with a positive serology in the first trimester, remained positive in the third trimester. This result highlights that naturally acquired immunity against SARS-CoV2 may last for several months.

### 4.2. Comparison with Previous Studies

We previously demonstrated that, in Madrid region, rate of SARS-CoV2 infection among pregnant women was similar to that reported in the general population [7]. A nationwide, population-based sero-epidemiological study carried out between 27 April and 11 May 2020, including 51,958 samples obtained from all Spanish regions, reported an overall SARS-CoV2 seroprevalence of 4.6%. However, there was a geographical variability, and Madrid showed a much higher seroprevalence rate of 11.5% [12] which is concordant with our findings.

Despite the growing number of published articles, only a few studies have evaluated the seroprevalence of SARS-CoV2 infection at different stages of pregnancy during the 2020 outbreak of COVID-19 in Spain. The reported prevalence of positive serological tests in pregnant women in our country varied from 15% in the first trimester [13], to 20% in the third trimester and delivery [13,14,15,16]. There is a smaller-scale study, carried out at three hospitals in New York, involving 149 women who were assessed for anti-SARS-CoV2 IgG antibodies during the first and second trimesters, as well as at the moment of delivery [17]. The outcomes from this study were similar to our own findings as the authors reported a seroprevalence rate of 12.1% during the first trimester and 16.1% during the second trimester. Notably, 71.4% of the women who tested positive during the first trimester remained positive at the time of delivery, which is similar to the 66.7% observed in our cohort. The notable increase in seroconversion rates during the third trimester compared to the first trimester might be explained by maternal immunological changes occurring in the latter stages of pregnancy. These adaptations potentially heighten susceptibility to certain infections [5,18,19]. However, in our study, this shift could be linked to the timing of the first trimester, which coincided with the pandemic’s onset. Stringent preventive measures, especially for vulnerable groups like pregnant women, were rigorously implemented. As these measures eased during the 2020 summer, marking the end of extreme social isolation, a subsequent surge in COVID-19 cases occurred during the second wave, affecting the general population, including pregnant women.

Regarding obstetric and maternal morbidity, we expected a similar effect in pregnancy as that reported in the literature during previous pandemics (MERS, SARS-CoV-1 and influenza), with a higher rate of miscarriage when the infection took place in the first trimester of pregnancy and more cases of fetal growth restriction (FGR) when the infection occurred in late pregnancy [3,4,5]. However, the existing literature on obstetric morbidity among SARS-CoV-2-infected pregnant individuals presents conflicting findings demonstrating that our current understanding of COVID-19 infection across pregnancy trimesters remains limited. On the one hand and consistent with our results, certain authors have not identified statistically significant differences in obstetric complications. Cosma et al. did not observe elevated rates of early pregnancy loss in women infected with SARS-CoV-2 during their first trimester [20]. Similarly, Villalaín et al. and Juan et al. did not report an increased risk of adverse pregnancy outcomes such as FGR, preterm birth, or preeclampsia [21,22]. Conversely, numerous other studies have demonstrated higher rates of obstetric complications associated with SARS-CoV-2 infection, including preterm birth, premature rupture of membranes, low birth weight, and stillbirth [23,24,25,26].

Additionally, the current knowledge of COVID-19 infection in different trimesters of pregnancy is still limited. While efforts to compare obstetric morbidity based on the infection trimester have increased, there is ongoing debate. Several studies suggest a higher incidence of adverse fetal outcomes, including stillbirth, perinatal and neonatal death, and preterm birth in women infected during their first trimester. In contrast, infections in the third trimester seem associated with lower fetal growth percentile and higher rates of small for gestational age (SGA) fetuses [26,27].

However, there are still insufficient data assessing immunological status throughout pregnancy and, normally, only acute infection using rRT-PCR SARS-CoV2 has been assessed at a single time point. This could be leading to a selection bias, as the majority of the SARS-CoV2-infected population is actually asymptomatic and, therefore, no rRT-PCR will have been performed [5]. Di Mascio et al. [26] analyzed 388 pregnancies that had a positive rRT-PCR SARS-CoV2 test during pregnancy, describing how perinatal outcomes (stillbirth, perinatal and neonatal death, and preterm birth) were significantly worse with decreasing gestational age at the time of infection. In a retrospective study evaluating 882 positive pregnant women with rRT-PCR SARS-CoV2, including 85 women diagnosed in the first trimester, it was reported that gestational age at the time of infection was the best predictor for gestational age at delivery [27]. To the best of our knowledge, there is only one study that has been conducted to assess SARS-CoV2 serology during both the first and third trimesters while examining the potential association between the presence of antibodies and pregnancy outcomes [28]. In this study, which involved 528 singleton pregnant women, the authors carried out serological assessments during the initial 11–13-week screening visit and again upon the admission for delivery. Data from pregnancy outcomes (gestational age at delivery, preterm birth before 34 weeks, hypertensive disorders, gestational diabetes, and abnormal fetal growth) were exhaustively collected to investigate the association between obstetric morbidity and SASRS-CoV2 infection. They did not discover any significant association between serological status and major obstetric complications. It must be pointed out that our study conducted a serological analysis in the third trimester of pregnancy, at 35–36 weeks, which likely provides a more comprehensive evaluation of newly emerging complications.

The high virulence of this novel SARS-CoV2, coupled with the still ongoing debate about its potential adverse effects on pregnancy, underscores the importance of continued scientific research. Researchers should remain committed to exploring existing data in order to be better prepared for potential future viral threats and to provide the pregnant population with a specific and high-quality assessment and healthcare. Serological SARS-CoV2 screening stands as a valuable tool that can provide high quality evidence regarding the natural progression of the disease, its severity and prognosis based on the timing of infection, enhancing the clinical management of those infected women. Nevertheless, we want to highlight that various other variables could contribute to the susceptibility to SARS-CoV-2 infection. Factors such as the employment status of pregnant individuals—whether on maternity or pregnancy leave—and the nature of their careers (onsite vs. remote work), along with the size of their household or the number of children in a family, are crucial elements that could significantly impact susceptibility to SARS-CoV-2 infection. Recognizing these social aspects as influential factors underscores the necessity for proactive investigation and their inclusion in research endeavors. By integrating these social dimensions, future studies can better equip us to confront and prevent infections in potential future pandemics.

### 4.3. Strengths and Limitations

The main strength of our study is the longitudinal follow-up of a consecutive sample of pregnant women who were in their first trimester of pregnancy during the COVID-19 first outbreak in Madrid, Spain, one of the most severely affected countries in Europe at that time. This allowed us to analyze two blood samples, corresponding to the first and third trimesters, coinciding with the first and second waves of the COVID-19 pandemic. Assessing anti-SARS-CoV-2 immunoglobulins in these distinct pregnancy phases provided a unique opportunity to enhance our understanding of maternal immune response adaptations, explore the impact of social security measures on COVID-19 incidence, and investigate the relevance of infection timing on obstetric outcomes.

However, there are some limitations in our work that warrant acknowledgment. We consider that the main limitation of our study relates to the small sample size of our cohort, which might be responsible for the lack of statistically significant differences in the results. Consequently, this has prevented us to perform any subgroup analysis. In addition, we did not record individual measures to prevent SARS-CoV2 infection, therefore we have assumed that women were compliant with governmental restrictions implemented during this period of the COVID-19 pandemic. Another possible limitation found in our study is the lack of information concerning social characteristics such as working routines and family members, including children living in the same residence, which could have been useful to better analyze both groups, as it might have an impact on the predisposition to suffer from SARS-CoV2 infection.

## 5. Conclusions

The main conclusion drawn in the present study is that the COVID-19 seroconversion rate was higher in third than in the first trimester of pregnancy, covering the first and second waves of the COVID-19 pandemic, with the majority of the women infected during their first trimester remaining positive throughout gestation. The prompt implementation of SARS-CoV2 serological testing as part of the protocol in obstetric outpatient services in every trimester rutinary analysis would be able to detect asymptomatic cases and reflect an accurate COVID-19 seroconversion rate. While our study did not find any statically significant differences in maternal or obstetric complications based on the trimester of infection, larger studies, including social variables are still needed. Such studies are necessary to enhance preparedness for potential future viral threats and to mitigate the risk of disease contraction.

## Figures and Tables

**Figure 1 viruses-15-02386-f001:**
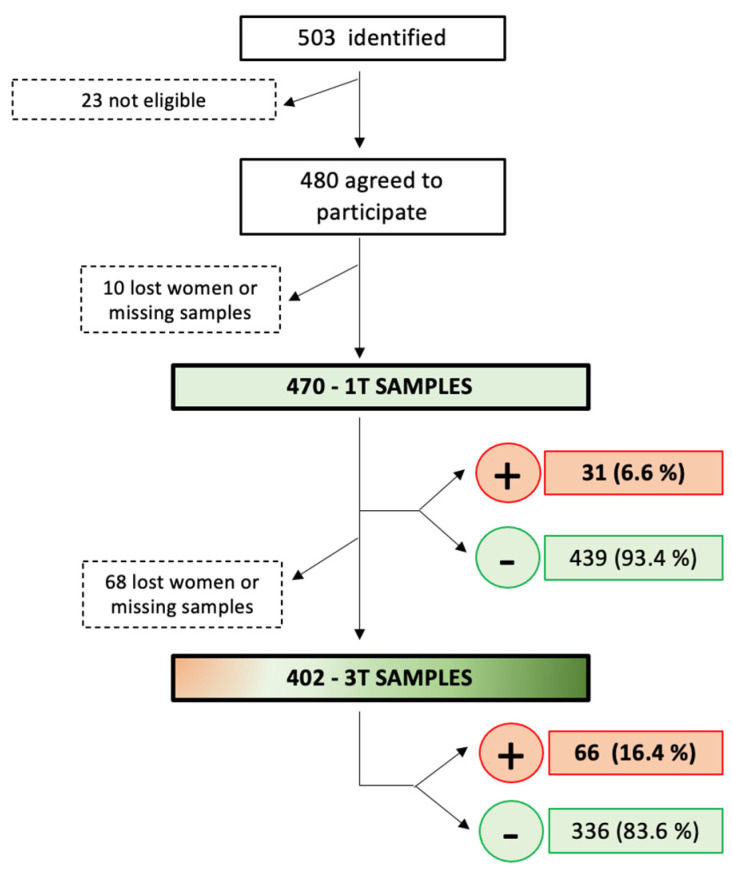
Patients’ flowchart, study process, and COVID-19 seroprevalence rate in first and third trimesters of pregnancy. 1T: first trimester; 3T: third trimester.

**Figure 2 viruses-15-02386-f002:**
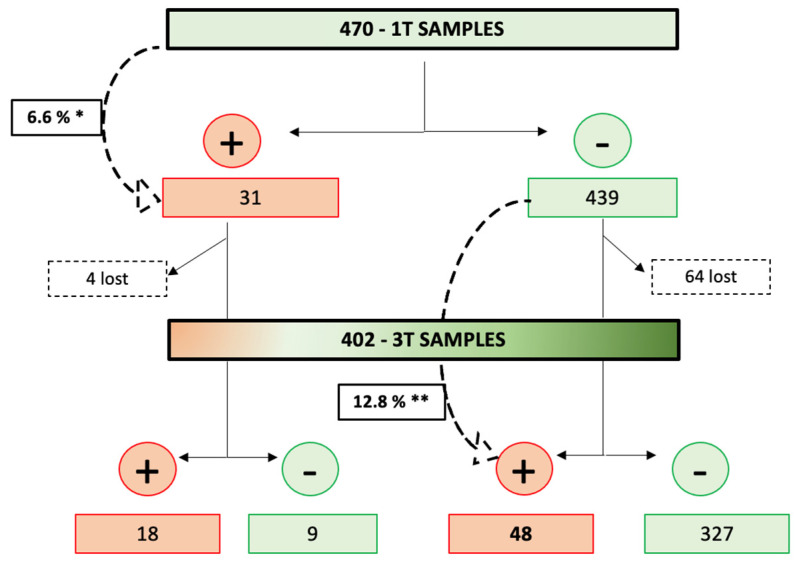
COVID-19 seroconversion rate throughout the pregnancies. 1T: first trimester; 3T: third trimester. * First-trimester seroconversion rate. ** Third-trimester seroconversion rate.

**Table 1 viruses-15-02386-t001:** Maternal baseline characteristics.

Variable	Positive Serology 1T (*n* = 31)	Positive Serology 3T (*n* = 48)	*p*-Value
Gestational age at delivery (weeks)	39.6 [38.9; 40.0] Missing: 2 (6.5%)	40.0 [38.6; 40.5] Missing: 0 (0%)	0.285
Maternal age (years)	35.0 [29.0; 38.0]	33.5 [29.8; 36.3]	0.398
Body mass index (kg/m^2^)	22.9 [21.5; 26.4]	25.3 [22.8; 28.6]	0.110
Nulliparous	17 (54.8%)	20 (41.7%)	0.356
Race			
Black	1 (3.2%)	2 (4.2%)	1.000
Non-Hispanic White	22 (71.0%)	37 (77.1%)	0.601
Hispanic/Latin	7 (22.6%)	6 (12.5%)	0.352
Asian	0	1 (2.1%)	1.000
North African	1 (3.2%)	1 (2.1%)	1.000
Other	0	1 (2.1%)	1.000
Blood type			
A Positive	12 (38.7%)	15 (31.3%)	0.628
A Negative	1 (3.2%)	1 (2.1%)	1.000
O Positive	15 (48.4%)	21 (43.8%)	0.818
O Negative	1 (3.2%)	3 (6.3%)	1.000
B Positive	2 (6.5%)	6 (12.5%)	0.470
B Negative	0	0	1.000
AB Positive	0	1 (2.1%)	1.000
AB Negative	0	1 (2.1%)	1.000
Active smoking	0	5 (10.4%)	0.151
Chronic medical pathology			
None	24 (77.4%)	33 (68.8%)	0.451
Hypertensive disorders	1 (3.2%)	0	0.392
Diabetes mellitus	0	0	1000
Autoimmune or Immunological disorders	2 (6.5%)	0	0.151
Respiratory disease	1 (3.2%)	2 (4.2%)	1.000
Others	7 (22.6%)	15 (31.3%)	0.451

1T: first trimester; 3T: third trimester.

**Table 2 viruses-15-02386-t002:** Obstetric complications according to serology group.

	Positive Serology 1T (*n* 30 *)	Positive Serology 3T (*n* 48)	*p*-Value
None	23 (76.7%)	33 (68.8%)	0.606
Gestational hypertension	1 (3.4%)	0	0.385
Preeclampsia	0	0	1.000
Gestational diabetes	3 (10.0%)	3 (6.3%)	0.670
Preterm birth	1 (3.3%)	2 (4.2%)	1.000
Other (cholestasis, Rh isoimmunization, shortened cervix, plaquetopenia…)	0	1 (2.1%)	1.000
Small for gestational age	0	4 (8.3%)	0.156
Fetal growth restriction	1 (3.3%)	3 (6.3%)	1.000
Fetal anomalies	1 (3.3%)	2 (4.2%)	1.000

* *n* = 30, as one woman was excluded from the positive serology 1T group (initially *n* = 31) due to early miscarriage in her first trimester of pregnancy. 1T: first trimester; 3T: third trimester; Rh: Rhesus.

## Data Availability

The data presented in this study are available upon request from the corresponding author. The data are not publicly available due to data protection regulations.

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
