# Peer review of "Maternal COVID-19 Serological Changes—Comparison between Seroconversion Rate in First and Third Trimesters of Pregnancy and Subsequent Obstetric Complications: A Cohort Study"

_viruses, 2023, doi:10.3390/v15122386_

Round 1

Reviewer 1 Report (Previous Reviewer 1)

Comments and Suggestions for Authors

1. Could the Authors specify the treatment of SARS-CoV-2 infection and vaccination status in included patients ? Did they have any comorbidities ?

2. I cannot see the mentioned supplementary material – can the Authors add the Figure S1: title; Table S1: title; Video S1: title ?

I sincerely appreciate the effort of the Authors to provide the new information of SARS-CoV-2 infection and its management especially in pregnant women, as I consider this infection still as the life-threatening complication for each patient and have the respect against it.

Thus, I recommend minor revision of the paper that might be published after the implementation of the responses to the comments of the reviewers in the manuscript.

Author Response

Thank you for considering our paper for publication in Viruses Journal. We appreciate the reviewer’s comments.

We include a revised manuscript and responses to the reviewers.

Reviewer 2 Report (New Reviewer)

Comments and Suggestions for Authors

I believe that further important information to add to the study and which could influence the different susceptibility to infections (in addition to whether the samples from the 1T group were collected during the lockdown) is also whether the women were working or if they were on maternity leave and the ages of other children. In conclusion, the authors should describe the social habits of the two groups. There are many variables that could influence what is observed. Furthermore, there is no discussion on the different abilities, if any, to produce an antibody response between the two groups to Sars-CoV-2.

Author Response

Thank you for considering our paper for publication in Viruses Journal. We appreciate the reviewer’s comments.

We include a revised manuscript and responses to the reviewers.

Round 2

Reviewer 2 Report (New Reviewer)

Comments and Suggestions for Authors

The authors have satisfactorily edited the manuscript following my instructions, therefore I endorse it for publication

This manuscript is a resubmission of an earlier submission. The following is a list of the peer review reports and author responses from that submission.

Round 1

Reviewer 1 Report

Comments and Suggestions for Authors

I highly appreciate the originality of the study and its usefulness for the clinical practice because of the risk of this infections for the general population with the potential to cause life-threatening respiratory insufficiency aggravated especially in the third trimester by the worsened mobility of the pregnant woman and her diaphragm by the fetus.

Content suggestions:

1.            Do the authors have any information about the personal history of the patients regarding immune-mediated diseases, immunodeficiency...?

2.       Do they have the data about the drugs of the included women – any medication influencing the immunity including vitamins ?

3.        I wonder if the authors know about the allergologic history of the patients ? It could influence the immunological status...?

4.    I would like to kindly ask them also to describe the treatment and recommendation of the management of COVID-19 infection in the included women. It could be helpful for the practice in general.

Taking the comments into account, I recommend minor revision of the paper and would be very happy to see it published it. The study is very important and after the implementation of the responses to the comments, it could improve the health of the woman and the newborn, as well.